# Changes of Thermostability, Organic Solvent, and pH Stability in *Geobacillus zalihae* HT1 and Its Mutant by Calcium Ion

**DOI:** 10.3390/ijms20102561

**Published:** 2019-05-24

**Authors:** Siti Nor Hasmah Ishak, Malihe Masomian, Nor Hafizah Ahmad Kamarudin, Mohd Shukuri Mohamad Ali, Thean Chor Leow, Raja Noor Zaliha Raja Abd. Rahman

**Affiliations:** 1Enzyme and Microbial Technology Research Centre, Faculty of Biotechnology and Biomolecular Sciences, Universiti Putra Malaysia, 43400 UPM Serdang, Selangor, Malaysia; snhasmahishak@gmail.com (S.N.H.I.); hafizah_kamar@upm.edu.my (N.H.A.K.); mshukuri@upm.edu.my (M.S.M.A.); adamleow@upm.edu.my (T.C.L.); 2Department of Microbiology, Faculty of Biotechnology and Biomolecular Sciences, Universiti Putra Malaysia, 43400 UPM Serdang, Selangor, Malaysia; 3Centre for Virus and Vaccine Research, School of Science and Technology, Sunway University, Bandar Sunway 47500 Kuala Lumpur, Selangor, Malaysia; malihem@sunway.edu.my; 4Centre of Foundation Studies for Agricultural Science, Universiti Putra Malaysia, 43400 UPM Serdang, Selangor, Malaysia; 5Department of Biochemistry, Faculty of Biotechnology and Biomolecular Sciences, Universiti Putra Malaysia, 43400 UPM Serdang, Selangor, Malaysia; 6Department of Cell and Molecular Biology, Faculty of Biotechnology and Biomolecular Sciences, Universiti Putra Malaysia, 43400 UPM Serdang, Selangor, Malaysia; 7Institute of Bioscience, Universiti Putra Malaysia, 43400 UPM Serdang, Selangor, Malaysia; 8Laboratory of Halal Science Research, Halal Products Research Institute, Universiti Putra Malaysia, 43400 UPM Serdang, Selangor, Malaysia

**Keywords:** T1 lipase, Calcium ion, metal-binding site, *Geobacillus zalihae*, thermostability, organic solvent tolerance

## Abstract

Thermostable T1 lipase from *Geobacillus zalihae* has been crystallized using counter-diffusion method under space and Earth conditions. The comparison of the three-dimensional structures from both crystallized proteins show differences in the formation of hydrogen bond and ion interactions. Hydrogen bond and ion interaction are important in the stabilization of protein structure towards extreme temperature and organic solvents. In this study, the differences of hydrogen bond interactions at position Asp43, Thr118, Glu250, and Asn304 and ion interaction at position Glu226 was chosen to imitate space-grown crystal structure, and the impact of these combined interactions in T1 lipase-mutated structure was studied. Using space-grown T1 lipase structure as a reference, subsequent simultaneous mutation D43E, T118N, E226D, E250L, and N304E was performed on recombinant wild-type T1 lipase (wt-HT1) to generate a quintuple mutant term as 5M mutant lipase. This mutant lipase shared similar characteristics to its wild-type in terms of optimal pH and temperature. The stability of mutant 5M lipase improved significantly in acidic and alkaline pH as compared to wt-HT1. 5M lipase was highly stable in organic solvents such as dimethyl sulfoxide (DMSO), methanol, and n-hexane compared to wt-HT1. Both wild-type and mutant lipases were found highly activated in calcium as compared to other metal ions due to the presence of calcium-binding site for thermostability. The presence of calcium prolonged the half-life of mutant 5M and wt-HT1, and at the same time increased their melting temperature (T_m_). The melting temperature of 5M and wt-HT1 lipases increased at 8.4 and 12.1 °C, respectively, in the presence of calcium as compared to those without. Calcium enhanced the stability of mutant 5M in 25% (*v*/*v*) DMSO, n-hexane, and n-heptane. The lipase activity of wt-HT1 also increased in 25% (*v*/*v*) ethanol, methanol, acetonitrile, n-hexane, and n-heptane in the presence of calcium. The current study showed that the accumulation of amino acid substitutions D43E, T118N, E226D, E250L, and N304E produced highly stable T1 mutant when hydrolyzing oil in selected organic solvents such as DMSO, n-hexane, and n-heptane. It is also believed that calcium ion plays important role in regulating lipase thermostability.

## 1. Introduction

Over the past decade, protein crystallization under space condition has attracted special attention among researchers. The lack of sedimentation and reduction of buoyancy-driven convection in space provide an ideal condition for protein crystal growth. Beside the improvement of size, perfection, and morphology of crystals, the space condition also offers a perfect environment for the formation of high-quality protein crystal with better packing arrangements. This type of protein crystal produced in space environment leads to the production of high-quality diffraction patterns during X-ray diffraction analysis, allowing substantially more precise and explicit information and translation of its protein structure compared to Earth-grown crystals [1]. The quality of the crystal is important, as it comprises the details of the electron density map and allows for better understanding of the mechanism and structural functions of proteins. As reported, protein crystal derived from space condition provides high-quality diffraction data resulting in a precise model structure of whale myoglobin triple mutant mb-YQR [1]. The comparisons of structural data from space- and Earth-grown crystal structures of acidic phospholipase A2 concluded that the space environment does not modify the polypeptide chains of protein; however, it showed some improvement in the bound water molecules at the hydration layer [2]. Since the fluid convection-driven motion around the crystallizing protein depends on the level of gravity, the total number of bound water molecules may concurrently vary, hereby influencing the number of interactions in the structure.

The molecular dynamics (MD) simulation approach is widely applied in the discovery of structure-function relationship and molecular arrangements in protein structure. This approach has been also used to explore the behavior of molecules under certain conditions, which are impossible to achieve in the laboratory. MD simulations were extensively used to investigate the conformational transition and molecular mechanism of macromolecules. The analysis of root mean square deviation (RMSD), root mean square fluctuation (RMSF), and radius of gyration obtained from analysis of protein structure using MD simulation provides information on the fluctuations and conformational changes of macromolecules. This information obtained from MD analysis could be applied to determine the overall stability of protein structure and secondary structure elements [3]. The capability of the MD simulation approach to differentiate the stability and quality of two similar models makes it possible to study the impact of mutation on structural changes in protein [4]. As reported, Kamaraj and Purohit [5] used in silico screening and MD simulation approach to select the most potential mutation that might be associated with Oculocutaneous Albinism type III (OCA3). This computational method was used to predict the structure rigidity of mutants that could increase the stability of lipase A from *Bacillus subtilis* [6]. Besides structural stability, the catalytic activity of lipase during amino acid substitution could be also determined using MD simulation approach [7]. By analyzing MD simulation results, the surface residues of *Candida antarctica* lipase B (CalB), which displayed high RMSD values in methanol solvent, were concluded as attenuated by methanol and selected for amino acid substitutions [8].

There are few contributing factors in maintaining the structure and catalytic role of enzymes under extreme conditions such as hydrogen bond formation, ion interaction, hydrophobic interaction, and interaction with metal ions [9,10,11]. Hydrogen bond and ion interaction are important to stabilize the folding structure of protein, hence retain their activity [12]. The number of hydrogen bonds in protein and types of hydrogen bond are related to the thermostability of protein, where different energy is produced by each interaction [13,14]. The introduction of hydrogen bonds in *Bacillus* lipase by amino acid mutation resulted in the increase of protein rigidity and thermostability [15]. The substitution of amino acid strategy resulted in additional non-covalent interactions also reported to increase the stability of enzyme against organic solvents. Hydrogen bonds could be important structural features for enzyme stability in the presence of a hydrophilic organic solvent. Hydrogen bond formation via rational engineering of *Candida antarctica* lipase B (CalB) at targeted amino acid residues of N97Q, N264Q, and D265E successfully increased mutated lipase stability in methanol compared with the wild-type [16]. The structural changes, such as the formation of ion pair interaction and hydrogen bonds developed by amino acids substitution, led to an improvement of the hydrophobic core and shift in the side chain isoelectronic point (pI) value, hereby improving the organic solvent stability of the protein [17]. The amino acid substitution approach successfully increased *Pseudomonas aeruginosa* LST-03 and *Bacillus subtilis* lipases in DMSO [17,18]. As a leading biocatalyst, it is important to have lipase with special characteristic to withstand the presence of various organic solvents. Lipase is capable of catalyzing various reactions, including hydrolysis, interesterification, alcoholysis, acidolysis, esterification, and aminolysis. The stability and flexibility of lipase to catalyze these reactions effectively under extreme conditions such as organic solvents, high temperatures, and salinity makes these enzymes highly attractive from a commercial perspective. Organic solvent-tolerant lipase is highly important in the application of fatty acid ester synthesis, biodiesel processes, as well as food and detergent industries [19]. Therefore, to achieve high demand for organic solvent-tolerant lipase, site-directed mutagenesis could be applied to improve the characteristics of the enzyme to increase the tolerance and stability for broad applications.

The presence of metal ions such as zinc and calcium ions is known to assist in enzymatic activity of some proteins such as lipases from *Staphylococcus xylosus* and *Geobacillus stearothermophilus* [10,20]. The presence of calcium ion enhances the catalytic activities of lipases from *Staphylococcus hyicus*, *Aeromonas sobria*, and *Geobacillus* sp. [21,22,23]. Metal ion is known to preserve and stabilize the structure of protein molecules, where the presence of calcium ion and LipB (specific modulating protein) was found essential in the formation of lipase-modulator complexes that reactivates the denatured protein, LipA lipase from *Pseudomonas euruginosa* [24]. X-ray crystallographic structure from *Pseudomonas glumae* lipase showed that calcium ion strongly binds to the protein structure and contributes to the overall structure stabilization following heat treatments and proteolysis. Previous studies identified lipases from *Staphylococcus hyicus* (SHL) and *Bacillus stearothermophilus* harboring a metal-binding site for folding stabilization at high temperature [21,25]. There is also different view on metal-binding effects in which Bertoldo et al. [11] indicated that the structure of thermostable lipase from *Staphylococcus xylosus* was not affected by the presence of calcium ion. Hitherto, the lack of information on the effects of calcium ions to the activity and stability of thermostable lipase at high temperatures remained to be debated among researchers. Several thermostable lipases such as lipases from *Bacillus* sp. and *S. hyicus* have been studied based on the influence of calcium to hydrolytic activity and stability [25,26]. 

A thermostable enzyme T1 lipase that has been successfully extracted from *Geobacillus zalihae* isolated from palm oil mill effluent in Malaysia showed its optimal lipase activity at 70 °C and was very stable at alkaline pH [27]. Previously, T1 lipase recombinant consisted of GST-tag and required two steps of affinity chromatography for purification process. The replacement of GST-tag with His-tag in the recombinant plasmid consist of T1 lipase gene (HT1) provides a fast and coherent purification step which involved only one-step affinity chromatography [28]. The first crystal structure of T1 lipase (2DSN) was elucidated at a resolution of 1.5 Å [29]. Under the collaboration of Japan Aerospace Exploration Agency (JAXA) and Universiti Putra Malaysia (UPM) Protein Crystal Growth (PCG) #2 Flight Program, T1 lipase protein was sent to the International Space Station (ISS) for crystallization process using counter-diffusion method and crystallized in a laboratory on Earth. Space-grown and Earth-grown crystal structures of T1 lipase were successfully elucidated at resolution 1.1 and 1.3 Å, respectively [30]. The comparison of both structures showed that space condition does not affect the amino acid sequence of lipase, but changes the structure arrangements of protein structure. The analysis of MD simulation study of space- and Earth-grown crystal structure of T1 lipase showed that space-grown crystal structure displayed low value of radius of gyration, RMSD, and RMSF, indicating the improvement of structural stability of the space-grown structure. The increasing value of RMSD, RMSF, and radius of gyration in Earth-grown structure of T1 lipase due to the enhanced motion, atomic mobility, and flexibility, suggesting that Earth-grown structure has lower structural stability compared to space-grown structure [31]. Space-grown crystal structure also displayed high average of hydrogen bond and ion pair interaction in its structure, which are important in structure stability. Earth-grown structure was found missing in interaction between residues Asp43–Gln39, Thr118–Gln59, Glu226–Gln254, Glu250–Arg330, and Asn304–Thr306 in its structure. The Earth-grown structure was also reported to possess an extra ion pair interaction between residues Glu250–Arg330 [31]. Crystal structure analysis of T1 lipase revealed the existence of calcium ion-binding site at Asp365, Glu360, Gly286, and Pro366 [32]. Through this observation, the function of calcium ion towards changes of activity and stability in T1 lipase are suggested to be highly adamant. Using space-grown crystal structure as a reference, a multiple site-directed mutagenesis was conducted on recombinant T1 lipase from *Geobacillus zalihae* to generate additional hydrogen bond and ion pair interaction. This is the first study involved in the manipulation of structural information obtained from space-grown crystal structure for the production of new enzyme to study the activity and stability against various organic solvents and elevated temperatures. Therefore, in the present work, the effect of mutations targeting on the residue involved in promoting new formation of hydrogen bond and ion pair interaction were investigated through lipase activity and stability, while incorporating calcium as part of a key component to influence lipase stability, especially in organic solvents. 

## 2. Results 

### 2.1. Construction of Mutant Lipase and Purification

The previous investigation explained the presence of additional hydrogen bonds and ion interactions in space-grown crystal structure. The MD simulation study of both Earth- and space-grown crystal structures suggested that ionic interaction in space-grown structure are much stronger [31]. Based on the data obtained from the earlier study by Ishak et al. [31], suitable amino acids were determined to establish new formation of hydrogen bond and ion interaction. Substitution of amino acids D43E, T118N, and N304E were chosen to add hydrogen bonds with amino acids Gln39, Asn59, Thr306, respectively, and E226D was selected for both hydrogen bond and ion interaction study with Arg230. Meanwhile, mutation of E250L was selected to add hydrogen bond with amino acid Gln254 and to break-ion interaction with Arg330, which is found missing in space-grown T1 lipase crystal structure. The modified recombinant plasmid, designated as modified PGEX-4T1/His, was used for the expression of both wild-type T1 lipase enzyme and mutant. T1 lipase enzyme (wt-HT1) used in this study consists of his-tag at the N-terminal which differentiated this T1 lipase from the previous study by Leow et al. [27]. The sequencing result confirmed the simultaneous amino acid substitution at positions 43, 118, 226, 250, and 304 named as 5M mutant lipase (Figure 1). Multiple sequence alignment between wt-HT1 and mutant 5M lipase displayed nucleotide changes at desired locations (Appendix A). Each recombinant wt-HT1 and 5M mutant lipase were purified using Ni Sepharose affinity chromatography with a final yield of 56.3% and 55.3% for wt-HT1 and 5M mutant, respectively. The specific activity of purified lipase from wt-HT1 was 19.1 and 18.7 U/mg for 5M mutant lipase, after purification. SDS-PAGE confirmed the presence of a single band approximately 44 kDa representing the wt-HT1 and 5M mutant lipase with 100 µg of each sample loaded into the well (Figure 2). 

### 2.2. Optimum Temperature and Thermostability of Lipases

The biochemical characterizations of wild-type (wt-HT1) and mutant (5M) lipases were performed under similar conditions. Both wt-HT1 and 5M mutant lipase displayed a comparable optimum temperature at 70 °C for lipase activity (Figure 3a). The thermostability study of mutant and wild-type lipases at broad temperatures are shown in Figure 3b. Results indicated that the two lipases were stable when incubated at temperatures starting from 30–60 °C for 30 min. The relative activity of 5M lipase was reduced to 70% at 70 °C when compared to wt-HT1. Both lipases were found to exhibit total activity loss at 90 °C. 

### 2.3. Optimum pH and pH Stability of Lipases

The effect of pH on changes of lipase activity was determined by measuring the activity of studied lipases in buffers with different pH. Based on the results obtained in Figure 4, wt-HT1 lipase displayed greater than 80% of relative activity at pH 7–9, whereas 5M mutant lipase showed higher than 80% of relative activity at pH 8–9. The maximum activity for both lipases were found to be at pH 9. Interestingly, 5M mutant lipase gained better stability at lower pH (pH 5 and 6) compared to wt-HT1. The activity of wt-HT1 was high in alkaline-based (pH 8–9) buffers but 5M mutant achieved greater stability at extremely alkaline medium with pH 11 where its residual activity was 30% higher as compared to wt-HT1, which experienced complete loss in its activity. 

### 2.4. Effect of Metal Ions on Lipase Stability

The influence of metal ions to lipase stability and activity can vary in its diversifying role of inhibition and stimulation (Table 1). Most thermostable lipases require metal ions in the form of Na^2+,^ Ca^2+^, Mg^2+^, Ni^2+^, Fe^3+^, Cu^2+^, and Zn^2+^ as co-factors for better functions and stability. This study showed the two lipases generally exhibiting an independent relationship between their hydrolytic activity and the types of metal ions. Metal ions such as Fe^3+^, Ni^2+^, Cu^2+^, and Zn^2+^ clearly inhibited lipase activities from wt-HT1 and 5M mutant. Magnesium (Mg^2+^) at 1 and 5 mM inhibited half of lipase activity in 5M mutant. As for wt-HT1, a high concentration of Mg^2+^ (5 mM), brought this lipase to better structure stability compared to the lower concentration (1 mM) of Mg^2+^ which down-regulated the activity at 20%. In addition, this study showed that sodium ions could retain the lipase activity of wt-HT1 at 100%. Mutant 5M however showed a down-regulation of lipase activity following increase in concentration of sodium ions. As a conclusion, mutant 5M and wt-HT1 are found to be of Ca^2+^-induced lipase. The addition of 5 mM calcium ions increased activity of 5M lipase by 190% while low concentration of calcium ions resulted in 150% of residual activity. At 1 mM and 5 mM calcium, the lipase activity of wt-HT1 enhanced by 20%.

### 2.5. Thermostability of Mutant in the Presence of Calcium Ion

The thermostability of wt-HT1 and 5M mutant lipases in calcium was determined via half-life study at 70 and 80 °C. Figure 5 shows that without calcium, mutant lipase 5M stability subsequently declined in which the lipase obtained a shorter period of the half-life 8 min when incubated at 80 °C and 23 min at 70 °C. Unlike 5M, wt-HT1 has a slight increase in its half-life with 9 h at 60 °C, 85 and 16 min, at 70 and 80 °C, respectively. These results strongly indicate the influence of multiple residue substitution on the level of thermostability showed by lipase 5M. From here, it was clear that the presence of 1 mM calcium in the reaction medium could bring improvements of lipase stability after incubation at these temperatures. Mutant lipase 5M showed its preference to calcium by extending its half-life to 40 min at 80 °C and 150 min at 70 °C. For wt-HT1 lipase, a significant surge in half-life was expected with 1 mM calcium where 50 % activity remained until 165 min at 80 °C and 250 min at 70 °C.

### 2.6. Effect of Organic Solvent and Surfactants in Lipase Activity in the Presence of Calcium Ions

The activity of lipases was inversely independent on the polarities of solvent with and without calcium. In the absence of calcium, both lipases activities did not change when being hydrolyzed in several organic solvents such as acetonitrile, 1-propanol, benzene, toluene, octanol, xylene, and n-heptane, except in DMSO, methanol, ethanol, and hexane, as presented in Figure 6. 5M lipase activity increased when being treated with 25% (*v*/*v*) DMSO and n-hexane at 133.5% and 120% of respected relative activity. The hydrolytic activity of 5M mutant lipase was moderately high in miscible solvent such as methanol and ethanol compared to wt-HT1. Meanwhile 25% (*v*/*v*) of 1-propanol inhibited the activity of 5M and wt-HT1 lipases to nearly 90%. Further addition of Ca^2+^ at 1 mM increased the residual activity of 5M and wt-HT1 lipases in methanol, ethanol, acetonitrile, n-hexane, and n-heptane. The lipase activity of mutant 5M increased as much as 2.7-fold in n-heptane and 1.2-fold in DMSO with calcium ion. Without calcium, the relative activity of 5M mutant in DMSO was found to be 20% lower. However, the presence of calcium ion was not affected by the activity of wt-HT1 lipase in the same solvent. On other hand, the enzyme activities slightly decreased in the presence of calcium ion in toluene, octanol, and xylene. Furthermore, the addition of Ca^2+^ did not affect the activity of enzymes in ethanol and 1-propanol. 

### 2.7. Secondary Structure and Melting Point Analysis

The effects of calcium on secondary structures of 5M mutant and wt-HT1 lipases were investigated by using circular dichroism (CD) spectroscopy analysis. Based on the findings, no significant alteration occurred in secondary structure of both wt-HT1 and 5M lipases. Instead, the CD spectra used to confirm the structure conformation of wt-HT1 and 5M lipases, which revealed a smooth asymptotic curve when scanned under a broad spectrum (190–240 nm) (Figure 7). The CD spectra results indicated that in the absence of calcium, higher percentage of α-helix was observed followed by lower percentage of β-sheet at 20 °C in both wt-HT1 and 5M mutant lipases (Table 2). However, the percentage of random structure of 5M lipase was slightly higher compared to the wt-HT1 at 39.7% and 31.3%, respectively. The percentage of α-helix in wt-HT1 was higher than 5M lipase at 28.7%. Both of lipases showed a declining α-helix ratio and increased of β-sheet ratio after being treated with calcium. As for the random coiled structure, the percentages significantly declined in the presence of calcium for both lipases. 

To understand the relationship between thermostability and lipase structure, protein denaturation study was initiated where the thermostability of wt-HT1 and 5M mutant lipases were determined by measuring the melting temperature (T_m_) from 30 to 95 °C. In Appendix A and S3, the spectral analysis of T_m_ obtained from wt-HT1 and 5M lipases with and without calcium ion were displayed clearly on the starting point of melting temperature in both lipases. These results suggested that the substitutions of five amino acids are the cause of the decline in T_m_ value of 5M lipase as described in Table 3, which follows the previous outcome of thermostability. The melting temperature of 5M mutant, which was reported at 67.6 °C, situates at 3.3 °C lower than wt-HT1 lipase (70.9 °C). However, with calcium the melting temperature of 5M mutant fluctuated to 76.0 °C while delaying further on the T_m_ of wt-HT1 lipase to 83.0 °C. 

### 2.8. Homology Modeling and Structural Analysis

A homology model of 5M mutant lipase was developed based on the available crystal structure of *G. zalihae* T1 lipase (Protein Data Bank code 2DSN) to improve understanding on mutagenesis effects on structure of lipase. The model structure of 5M was validated using RAMPAGE, ERRAT, and Verify-3D (data not shown). The superimposition of 5M model structure and Earth-grown T1 lipase structure gave the RMSD value of 1.37 Å (Figure 8). The 5M mutant model also showed the existence of sodium ion, which was not present in T1 lipase crystal structure. This ion was located near the catalytic site of the 5M mutant. Chloride ion-, zinc ion-, and calcium ion-binding sites were presented in both structures. Zinc binding site was coordinated by residues Asp61, His81, His87, and Asp238 in tetrahedral form as shown in T1 lipase crystal structure. Meanwhile, the 5M model showed that zinc ion was ligated by residues Asp61, Tyr77, His81, His87, and Asp238 (Figure 9a,b). The calcium-binding site for both structures was coordinated by residues Asp365, Glu360, Gly286, and Pro366 (Figure 9c,d). Based on the model structure, four of the mutations (D43E, E226D, E250L, and N304E) were located on the surface of the enzyme as shown in Figure 10. The T118N mutation was located at the helix inside the core of the protein structure, which is near to the catalytic sites. Mutation D43E is located at the first helix after the N-terminal. The amino acid substitution Asp to Glu successfully introduced a hydrogen bond between amino acid 43 and Gln39 as presented in Figure 11a. The substitution of amino acid Glu226Asp is located at helix 7, which is the lid of T1 lipase resulted in new formation of hydrogen bond with amino acid Arg230. The substitutions of amino acid E250L were subjected to break ion interaction with Arg330. As shown in Figure 11g, the substitution caused the loss of ion interaction between the substitution target residues and resulted in new formation of two hydrogen bonds between residues Arg330 and Leu307 located at the loop. In addition, new hydrogen bonds were also introduced between Leu250 and Gln254. The reason for the N304E mutation was to develop new formations of hydrogen bond with Thr306. However, the substitution caused the loss of hydrogen bonds with Leu307, and residue Leu307 was involved in the new interaction with amino acid Arg330. The establishments of new interactions due to amino acid substitutions on the surface of the 5M mutant might be the reason for stability improvement against organic solvents. 

## 3. Discussion

Site-directed mutagenesis approach is a common method used to improve thermostability in selected enzymes depending on the selected amino acid and the position of the mutation. Previously, T1 lipase space-grown crystal structure showed the presence of extra hydrogen bonds and stronger ionic interactions in its structure [31]. These non-covalent interactions have been reported to be important in protein stabilization and folding [33]. These interactions such as hydrogen bond, hydrophobic interaction, and ion interaction increased the rigidity of protein and essentially preserved catalytic active structure as well as protects unfolding of protein structure. According to Pace et al. [34], hydrogen bonds make a favorable contribution to protein stability and suggested that the peptide group hydrogen bonds contributed a relatively larger role in stability than the side chain hydrogen bond. Several studies involving re-engineering of ion interactions and hydrogen bonds through amino acid substitution in protein structures significantly improved the protein stability [35]. Previous reports indicated that the site-directed mutagenesis approach has successfully introduced additional hydrogen bonds and ion interactions in protein structure, which also contributed to the improvement of protein stability. The importance of hydrogen bond and ion interaction in protein stability and the impact of mutation on hydrogen bond formation has been elaborated upon in the studies. Substitution of H86Y/A269T in lipase of *Geobacillus stearothermophilus* via random mutagenesis resulted in the new formation of hydrogen bonds hence improving the stability of lipase [36]. Following these findings, five amino acids were chosen for site-directed mutagenesis. The substitutions of five amino acids in wt-HT1 lipase resulted in a similar pattern in optimum temperature and pH with its wild-type, HT1. Sharma et al. [37] reported that both wild-type (LipR1) and mutant (LipR1M1) from metagenome-derived lipase displayed similar profile in pH stability and pH optimum. In this study, focusing on the second-generation of mutant, 5M showed that this lipase has better stability towards acidic and alkaline pH (5 to 11). Analysis showed that 5M lipase retained almost 30% of its relative activity at pH 5 and pH 11 while wt-HT1 experienced lost in its activity at similar pH. The stability towards acidic pH can be attributed to the replacement of amino acid which has high pKa values that are capable of decreasing the pH in response to structure stability [38]. Other studies reported that wild-type LipR1 lipase showed higher activity towards alkaline pH values as compared to the mutated lipase LipR1M1, which was active at low pH. Other lipase variant (LipR4) displayed greater than 80% of activity at pH 7–9 with maximum activity at pH 8 [37,38]. The substitution of five amino acids in this study caused a decline in lipase stability when exposed to high temperature. The result also showed that 5M lipase exhibited low specific activity as compared to wt-HT1 lipase. This finding is rather similar to that happened in a mutant (S311C/R214C) where it showed a severe loss of lipase activity as compared to single mutation [24]. The mutation of S141G in *Pseudomonas fragi* lipase also had its rigidity relinquished, causing the structure to destabilize [39]. Therefore, amino acid substitution could successfully increase protein thermostability and has been reported by Veno et al. [40]. Through amino acid substitution of Gly210Cys in lipase derived from *Staphylococcus epidermidis,* the thermostability has been increased. Goomber et al. [15] reported that the substitution of amino acid Gln121 to Arg in LipJ lipase from *Bacillus* sp. resulted in a shift of optimal temperature from 37 to 50 °C apart from increasing the half-life of lipase from 20 min for wild-type to 2.5 h for variant at 50 °C. On the other hand, the substitution of amino acid Phe17 to Ser in lipase from *Bacillus thermocatenulatus* increased the relative activity of lipase at broad temperature (45 to 65 °C) but decreased in stability after 30 mins of incubation at 60 °C [41]. According to Singh et al. [42], the combination of two or more mutations in protein leads to either synergistic or antagonistic effects. Positive cooperation leads to synergistic effects, which improves the protein stability due to non-covalent interaction stabilization. Meanwhile antagonistic interactions can lead to structural destabilization. In this study, the combination of five amino acid substitutions does not significantly improve the protein stability.

Metal ion plays an important role in enzyme activity, as well as stability. In this study, metal ions such as Fe^3+,^ Cu^2+^ and Zn^2+^ potentially caused deterioration to lipase stability upon treatments with Fe^3+^, Cu^2+^ and Zn^2+^. The suppression of lipase activity in the presence of Fe^3+^ ion has been described by Khaleghinejad et al. [41], through the study of F16S mutation in *Bacillus thermocatenulatus* lipase. Mg^2+^ ion was found to slightly inhibit the activity of 5M and wt-HT1 lipase. In other studies, Ghori et al. [43] reported that Mg^2+^ and Fe^3+^ enhanced the activity of lipase from *Bacilus* sp. Interestingly, calcium was found able to enhance the stability of both 5M and wt-HT1 lipases following its increase in concentration. The addition of calcium ions also enhanced the activity of several lipases such as lipase from *Pseudomonas fluorescens* JCM5963, *Geobacillus stearothermophilus* JC and *Bacillus stearothermophilus* MC7 [44,45,46]. However, the activity of lipases from *Bacillus thermocatenulatus* and *Lactobacillus planetarium* were inhibited by the presence of calcium ions [41,47]. Based on the results obtained in our study, calcium provides positive implications to the stability and activity of studied lipases.

It is interesting to study the behavior of proteins in the presence of calcium ions. Rakesh et al. [38] suggested that calcium ions increased lipase activity through the action of calcium ion complexes on the released fatty acids. The binding of calcium ion to the enzyme molecules resulted in structure stabilization and enhanced the thermostability of 5M and wt-HT1 lipases. The half-life of both enzymes had increased by nearly five- and ten-fold at 80 °C, respectively. Kim et al. [25] showed similar results when L1 lipase thermostability decreased by about 10 °C in the absence of calcium ions, implying that calcium ions bound to the L1 lipase and stabilized its structure at higher temperature. The presence of calcium ions also enhanced the thermostability of *Pseudomonas aeruginosa* lipase, also important in catalytic activity [48]. Other calcium ion-dependent lipases reported were *Staphylococcus saprophyticus* lipase and KB-Lip from *Pseudomonas* sp. [49,50].

Lipases are widely used in industrial applications that involve organic solvents. Hence organic solvent tolerance is an important characteristic for industrial lipases. The stability of enzymes in organic solvents is crucial to ensure that the enzymes are active and functional in synthesis-based reactions. Lipase can encounter deactivation in water-free reaction due to different polarities exhibited by organic solvents. It is well known that the stability of enzyme involves several interactions such as hydrogen bonds, van der Waals forces, and hydrophobic interactions. The presence of water is important to maintain these interactions in enzymes. The exposure of enzymes to hydrophilic and polar solvents will destabilize the enzyme, since these solvents strip off the water from enzymes. Generally, enzymes are more stable in hydrophobic solvents since they possess less ability to remove the essential water from the enzyme molecule than hydrophilic solvents. Our results showed that substitution of amino acids increased the stability and activity of 5M lipase in DMSO with and without calcium ions. Li et al. [51] reported that amino acid substitution of *Serratia marcescens* lipase displayed various effects when treated in DMSO, with mutants G33D and T270A showing increment in activity while A187V was slightly lower in activity compared to wild-type. In contrast to our finding, about 20 % activity of *Pseudomonas fluorescens* lipase (rPFL) was lost after 24 hours incubation with DMSO [44]. Calcium ion has also reported diminished activity of lipase from *Chromohalobacter japonicus* BK-AB18 when treated with methanol, ethanol, and acetonitrile, although the presence of this cation significantly enhanced the activity of this enzyme in hexane [52]. In this study, the addition of calcium ion stimulated the activity of 5M mutant in ethanol, DMSO, n-hexane, and benzene. It is also enhanced the activity of wt-HT1 lipase in methanol, acetonitrile, ethanol, n-hexane, and n-heptane as compared with the enzymes without calcium ion.

It is known that the binding of metal ions may induce secondary structure changes in protein. It has been shown that calcium ion is important in catalytic activity of *Pseudomonas aeruginosa* lipase; however, this metal ion is able to promote reduction of α–helix in its secondary structure, hence decreasing enzyme stability [53]. The addition of calcium ion caused conformational changes in both wt-HT1 and 5M mutant lipases. The conformational changes of secondary structure elements in the presence of calcium ions has been reported in other proteins such as recombinant human bone morphogenetic protein 2 (rhBMP-2) with the increase of β-sheet and turn percentage as compared with proteins in the absence of calcium ion [54]. The formation of β-sheet structure in amphiphilic peptide increased sharply with respect to the unfolded structure with the addition of calcium ion [55]. 

The folding and unfolding state of a protein can be studied by CD to explore the conformational dynamics of protein. The analyses of melting temperature of 5M mutant and wt-HT1 lipases are in accordance with the thermostability study. The presence of calcium ion enhanced the thermostability of 5M mutant and wt-HT1 lipases by prolonging the enzyme’s half-life at 70 °C and 80 °C and increased the melting temperature. This explains that the calcium ion contributes to the stabilization of protein at high temperature in both structures. Similar findings were reported by Kim et al. [25], where the melting point of L1 lipase in the presence of calcium ion was 74 °C, which is 4 °C higher compared to the sample without calcium ion. The CD measurements can be used to verify the involvement of metal ions such as Ca^2+^ and Cd^2+^ in stabilization of protein secondary structure during unfolding process [56].

The improvement of protein stability in the presence of calcium ion is demonstrated by the presence of calcium ion-binding sites in both structures. However, the zinc ion-binding site presented in the protein structures does not assist in activity and stability as the addition of this element resulted in decreasing in enzymes stability. The substitution of amino acids resulted in a loss and gain of interactions where more new hydrogen bonds are established than lost. Even though the mutations of lipase did not improve the thermostability, it was found to increase in activity in the presence of DMSO. The substitution surface residues D43E, E226D, and E250L resulted in formations of new interactions, hence improved stability towards organic solvents, especially DMSO. It was previously reported that introduction of polar or charge amino acid residues and interactions between surface residues positively enhance enzyme stability in organic solvents. Substitution of residues H86Y and A269T, which are located at the enzyme surface of *Geobacillus stearothermophilus* Lipase T6 resulted in improved stability in the presence of methanol [36]. The hydrogen bond interactions with a tight bond of water molecules and the enzyme surface resulted in less water stripping. This indicates that the hydrogen bonding interaction between enzyme surface and water molecule is crucial to enzyme stability in organic solvents [16,36]. 

## 4. Materials and Methods 

### 4.1. Construction of Mutant Lipase 

Based on a previous study by Ishak et al., [31], five amino acids were selected to add hydrogen bond and ion pair interactions in T1 lipase structure. The in silico amino acid substitution was used to select the potential mutation for additional hydrogen bond and ion pair interaction. Hence five amino acids in T1 lipase were subjected to simultaneous mutation identified as D43E, T118N, E226D, E250L, and N304E. The oligonucleotide with desired mutations were synthesized by Bio Basic Inc. (Canada) and was cloned into plasmid pUC57. The gene harboring the original sequence of T1 lipase and five mutated residues were then double digested with restriction enzymes, EcoRI and BamHI before being ligated into modified pGEX-4T1 with His-tag [28]. Subsequently, the ligated plasmid was transformed into Top10 competent cell. To confirm the success of mutations, the plasmids from the few selected clones were sequenced by using the forward primer: 5′ CGG TGC ACC AAT GCT TCT GGC 3′ and reverse primer: 5′ GGG AGC TGC ATG TGT CAG AGG 3′. The plasmid that contained correct sequence of mutated lipase was transformed into expression vector BL21(DE3)pLysS. The expressed mutated lipase was later identified as 5M and unmutated T1 lipase was identified as wt-HT1. 

### 4.2. Protein Preparation and Purification

The *Escherichia coli* strain BL21(DE3)pLysS cells harboring the recombinant plasmid of 5M and wt-HT1 lipase were inoculated in 10 mL of Luria Bertani broth ( 1% (*v*/*w*) peptone, 1% (*v*/*w*) NaCl and 0.5% (*v*/*w*) yeast extract) supplemented with 100 μg/mL of ampicilin and 35 μg/mL of chloramphenicol and were grown at 37 °C overnight with shaking at 150 rpm. The resulting cultures were transferred into 200 mL of Luria Bertani broth and was induced with 0.025 mM IPTG when the OD600 reach 0.6–0.7 [12]. After 12 h of cultivation period at 37 °C and shaking at 150 rpm, the fermentation broth was centrifuged at 10,000× *g* at 4 °C for 30 min. The cell pellets were resuspended in binding buffer (20 mM phosphate buffer supplemented with 0.5 M NaCl and 0.5 mM imidazole, pH 7.4) prior to sonication. The sonicated protein was centrifuged at 10,000× *g* at 4 °C for 30 minutes and the supernatant was collected for one-step protein purification strategy using Nickel Sepharose affinity chromatography. The protein was eluted with 20 mM phosphate supplemented with 0.5 M NaCl and 0.5 M imidazole with gradient. 100 µg of each protein samples was loaded into the SDS-PAGE (SDS-polyacrylamide gel electrophoresis) gel. 

### 4.3. Lipase Assay and Protein Estimation

Protein concentrations were determined by Bradford assay [57] using the Bradford reagent from VWR Life Science AMRESCO (USA). Lipase activity was determined using modified Kwon and Rhee method [58]. Olive oil emulsion was prepared by emulsifying of olive oil (Bertoli, Italy) and buffer solution (1:1). The reaction mixture contained 10 μL enzyme, 990 μL buffer and 250 mL of emulsion. 20 μL of 20 mM calcium chloride were added prior to incubation at 70 °C for 30 minutes with 200 rpm shaking. Afterwards, 1 mL of 6N HCl was added to stop the reaction followed by 5 mL of isooctane. The reaction mixtures were left for 30 minutes at room temperature. The upper layer of the reaction was taken and mixed with 1 mL of copper pyridine solution pH 6.1. The activity was determined by measuring the absorbance at 715 nm. One unit (U) is the amount of enzyme that catalyzes the reaction of 1 μmol of substrate per minute. A calibration curve of oleic acid under the assay condition was used for enzyme activity calculating (Appendix A). The lipase activity was expressed as unit per mL (U/mL) according to the following formula:$$\mathrm{Lipase}\;\mathrm{Activity}\;(U/\mathrm{mL})=\frac{\left[\left(\Delta A_{715nm}-0.0193\right)/0.0212\right]}{\left(\mathrm{Reaction}\;\mathrm{Time}\times \mathrm{Enzyme}\;\mathrm{Volume}\right)}\;\Delta A_{715\mathrm{nm}}=A_{715\mathrm{nm}}\mathrm{sample}-A_{715\mathrm{nm}}\mathrm{blank}$$

### 4.4. Optimum Temperature and Thermostability

The optimum temperature of the 5M and wt-HT1 lipases were measured by assaying their hydrolytic activities on the olive oil emulsions at various temperatures ranging from 30 to 90 °C with 10 °C interval. The thermostability of enzymes were investigated by measuring the residual activity after pre-incubation at different temperatures (30 to 90 °C) for 30 mins followed by standard assay at their optimum temperature. The enzyme activity without pre-incubation was taken as 100 %. 

### 4.5. Optimum pH and pH Stability

The determination of pH optimum was studied by assaying the enzyme activity in different buffers with different pH values (50 mM sodium acetate buffer, pH 4–6; 50 mM sodium phosphate buffer, pH 6–7; 50 mM Tris-HCl, pH 7–9; 50 mM Glycine-NaOH buffer, pH 9–11). The pH stability was valued by pre-incubating the lipases in different pH buffers at 60 °C for 30 min prior to lipase assay. Subsequently, the remaining activity of the lipase measured according to the standard assay. 

### 4.6. Effect of Metal Ions on Lipase Activity

Effect of metal ions on lipase activity was evaluated by measuring the enzyme activity after pre-incubated the lipases at 60 °C for 30 min. Metal ions used in this study were Na^2+^, Ca^2+^, Mg^2+^, Ni^2+^, Fe^3+^, Cu^2+^ and Zn^2+^ with concentration of 1 and 5 mM. 

### 4.7. Thermostability of Mutant in the Presence of Calcium Ion

The half-life of the mutant at 70 and 80 °C was determined by pre-incubating the protein samples with and without 1 mM calcium ion for 5 and 3 h respectively prior to enzyme assay at optimum temperature. The samples were taken every 30- and 60-min intervals. The experiments were done in three replicates and the untreated enzymes activity was taken as 100%. 

### 4.8. Effect of Organic Solvents on Lipase Activity in the Presence of Calcium Ions

The effect of organic solvents on lipase stability were investigated with pre-incubated the lipases in 25% (*v*/*v*) of various organic solvents with and without the presence of 1 mM calcium ion. The solvents were selected based on their different log P values (values in parenthesis) and boiling point greater than 60 °C: Dimethyl sulfoxide (DMSO) (1.4), Methanol (−0.8), Acetonitrile (−0.4), Ethanol (−0.2), 1-propanol (0.3), Benzene (2.0), Toluene (2.5), Octanol (2.9), Xylene (3.1), N-hexane (3.5), and n-heptane (4.0). The mixture contains of 10 μL enzyme were pre-incubated at 60 °C for 30 minutes followed by standard lipase assay. 

### 4.9. Circular Dichroism Spectral Analysis

Secondary structure analysis and denaturation point (T_m_) of the proteins were measured with CD (Spectropolarimeter J-810, JASCO, Japan). The structural elements of wt-HT1 and 5M mutant lipases were estimated based on the far-UV spectral (190–240 nm) using 1 mm path cell length at temperature 20 °C. Samples contained purified protein at 1 mg/ml was prepared in buffer 5 mM Tris-HCl (pH 8.0). The concentration of 1 mM calcium ion was added into the protein solution to determine the structural elements in the presence of calcium ion. The spectral analysis of proteins was subtracted with the subsequent blank contained 5 mM Tris-HCl (pH 8.0) for analysis without calcium and 5 mM Tris-HCl (pH 8.0) supplemented with 1 mM calcium ions for analysis with calcium. For all measurements, a reference sample containing the corresponding buffer was subtracted from the CD signal. The preparation of protein was employed based on the protocol of Greenfield [59]. The analysis of structural element was determined using Spectra Manager^TM^ Suite Software (JASCO, Japan). 

The concentration of working enzyme was diluted to 1 mg/mL in 5 mM Tris-HCl pH 8 with and without 1 mM calcium ion. In this experiment, 1 cm optical path length was used, and the warming period was set from 30 to 90 °C for samples without calcium ion and 30 to 95 for samples contains calcium ion with a heating rate of 1 °C/min at 222 nm of wavelength. The denaturation of the protein investigated by collecting the complete spectra as a function of temperatures. The determination of melting temperatures of wt-HT1 and 5M mutant lipases were adopted based on the protocol of Greenfield [60]. The denaturation point (T_m_) of each sample was calculated using Spectra Manager^TM^ Suite Software (JASCO, Japan). The denaturation temperatures (T_m_) were defined as the point at which 50 % of the protein sample denatured. 

### 4.10. Homology Modeling and Structural Analysis

Yet Artificial Reality Application YASARA, version 10.2.1 [61] software was used to predict the structural model of 5M mutant lipase. The 5M mutant lipase was modeled via homology modeling using existing T1 lipase crystal structure (PDB ID: 2DSN) as template which has 99.3% sequence identity with 5M mutant. The modeled structures were validated using available online software such as RAMPAGE, ERRAT and Verify_3D tools [62,63,64]. The modeled structure of 5M mutant was compared and superimposed with Earth-grown T1 lipase crystal structure [30]. For hydrogen bond calculation, a donor-acceptor cutoff distance of 2.5 Å and acceptor-donor-hydrogen bond angle cutoff of 30° were considered. Ion interaction was determined when carbonyl atoms Asp or Glu side chain were found to be within 2.5 Å from nitrogen atoms of Arg, Lys, and His side chain. The visual analysis of structures and preparation of figures was carried out using YASARA software. 

## 5. Conclusions

Substitution of amino acid at five positions, D43E/T118N/E226D/E250L/N304E (5M mutant) lipase, enhanced the stability of the enzyme in DMSO, methanol, and n-hexane. Furthermore, 5M mutant lipase was also found to be more stable in acidic and alkaline pH. The presence of calcium ion dramatically increased the thermostability of lipases with the improvement of melting temperature and half-life of wt-HT1 and 5M mutant lipases. The addition of calcium ion also enhanced the stability of 5M lipase against various organic solvents such as DMSO, ethanol, n-hexane, and n-heptane. Meanwhile, it also improved the stability of wt-HT1 in ethanol, methanol, acetonitrile, n-hexane, and n-heptane. Calcium ion has been shown to be important in lipase thermostability. It would be interesting to investigate the impact of each mutation on protein stability by elucidating the crystal structure of 5M mutant lipase for a better understanding of the structure-function mechanism.

## Figures and Tables

**Figure 1 ijms-20-02561-f001:**
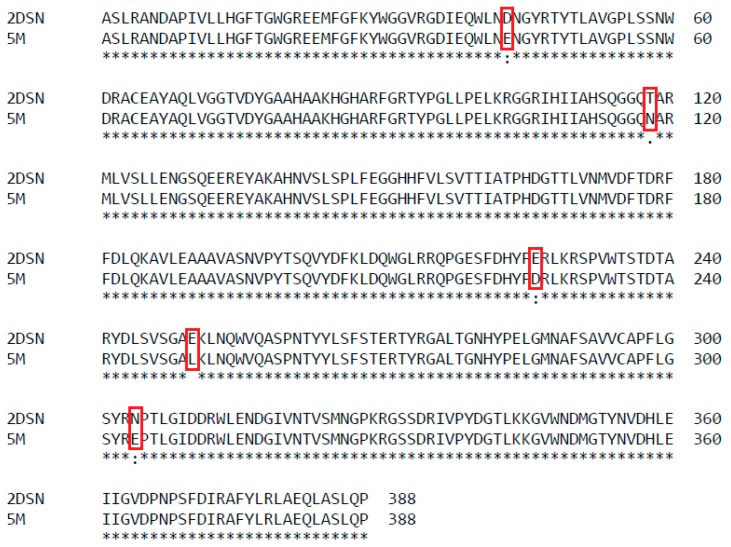
Alignment of 5M mutant with T1 lipase for verification of mutation at amino acid position 43, 118, 226, 250, and 304. The mutated residues are marked in red box.

**Figure 2 ijms-20-02561-f002:**
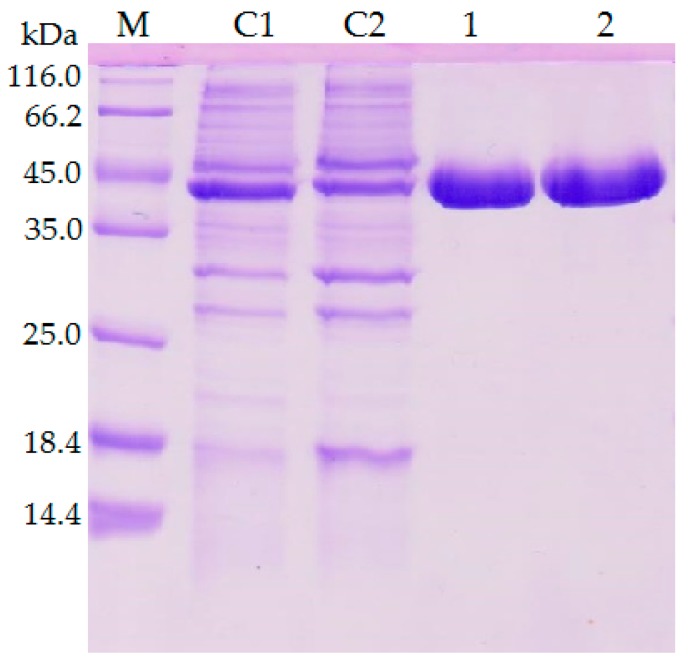
SDS-PAGE analysis of the wt-HT1 and 5M mutant lipases. The concentration of each samples (crude and purified enzymes) were standardized to 100 µg. Lane M; Molecular weight marker. Lane C1; crude enzyme of wt-HT1 lipase. Lane C2; crude enzyme of 5M mutant lipase. Lane 1; purified wt-HT1 lipase. Lane 2; purified 5M mutant lipase.

**Figure 3 ijms-20-02561-f003:**
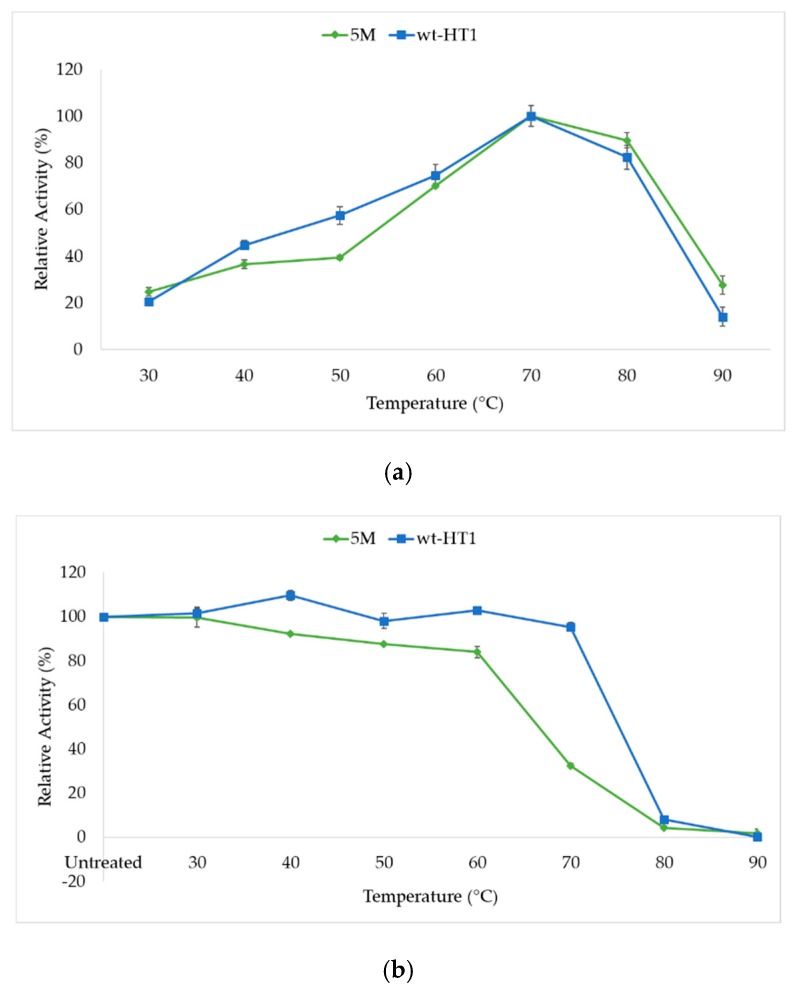
Effects of temperatures on activity and stability of lipases. (**a**) Optimum temperature of wt-HT1 and mutant 5M lipases. (**b**) Temperature stability of lipases from 30 to 90 °C.

**Figure 4 ijms-20-02561-f004:**
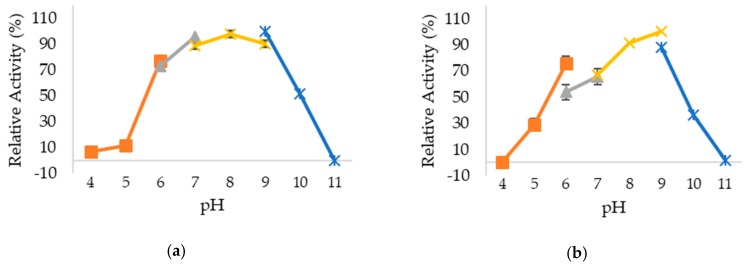
Optimum pH and stability of lipases in different pH values in buffer systems as follow; Sodium acetate buffer (pH 4–6), Sodium phosphate buffer (pH 6–7), Tris-HCl (pH 7–9) and Glycine-NaOH (pH 9–11). Sodium acetate buffer (pH 4–6), Sodium phosphate buffer (pH 6–7), Tris-HCl (pH 7–9) and Glycine-NaOH (pH 9–11). (**a**) Optimum pH of wt-HT1 lipase. (**b**) Optimum pH of 5M mutant lipase. (**c**) pH stability of wt-HT1 lipase. (**d**) pH stability of 5M mutant lipase.

**Figure 5 ijms-20-02561-f005:**
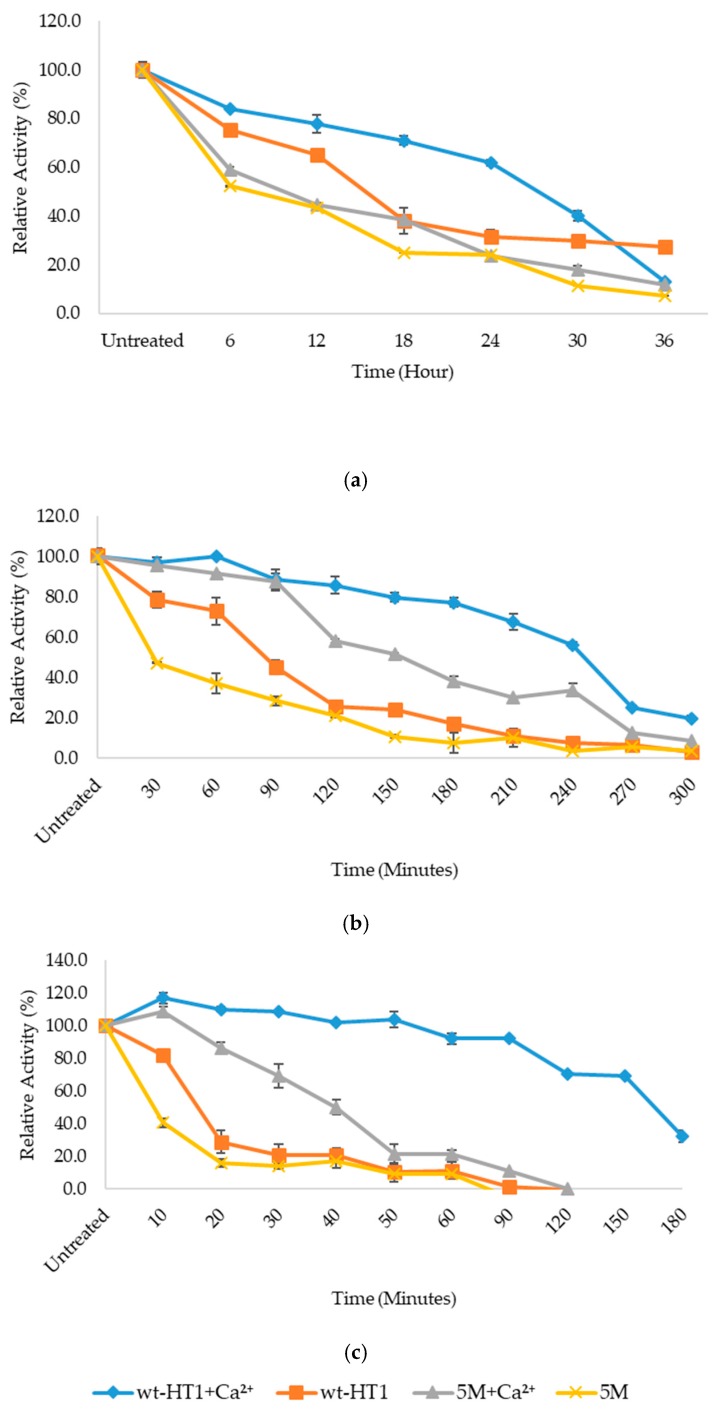
Thermostability of wt-HT1 and 5M mutant lipases. (**a**) Temperature profile at 60 °C. (**b**) Temperature profile at 70 °C. (**c**) Temperature profile at 80 °C.

**Figure 6 ijms-20-02561-f006:**
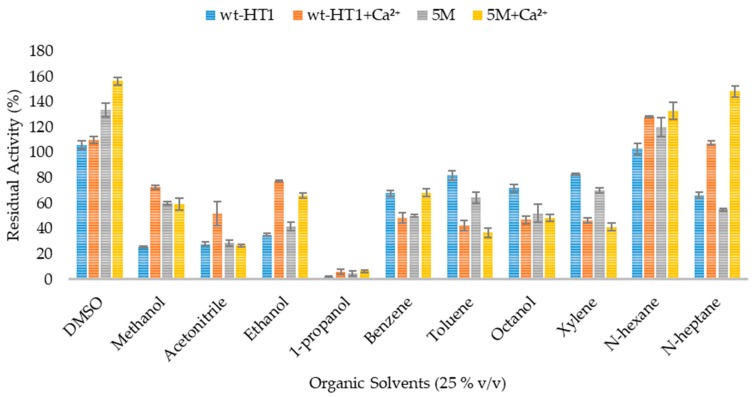
Effect of organic solvent on lipases activities with and without the presence of 1 mM Calcium ion.

**Figure 7 ijms-20-02561-f007:**
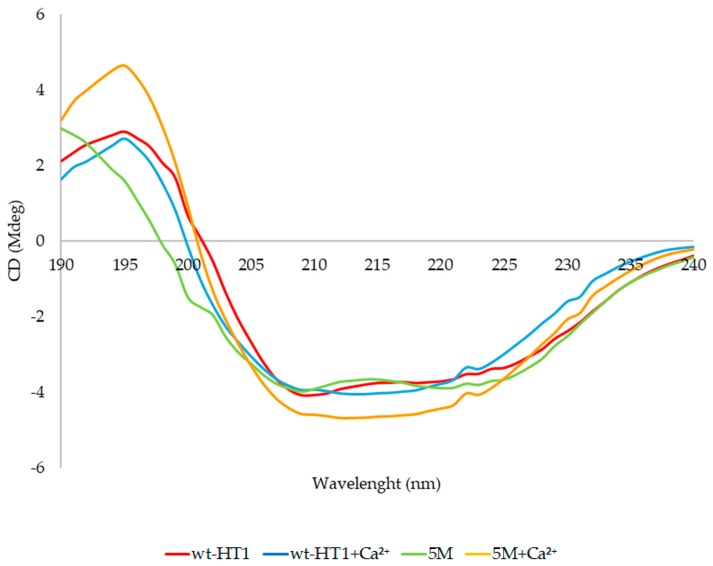
Conformational changes in the secondary structure of wt-HT1 and 5M mutant lipases at 20 °C.

**Figure 8 ijms-20-02561-f008:**
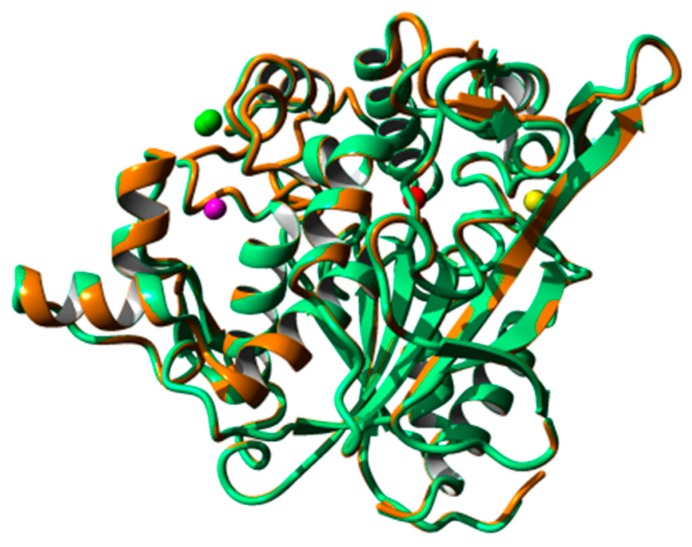
Superimposed of 5M mutant modeled structure (color in gold) with T1 lipase crystal structure (color in green). The calcium ion, zinc ion, sodium ion, and chloride ion shown as sphere color by yellow, magenta, red, and green, respectively.

**Figure 9 ijms-20-02561-f009:**
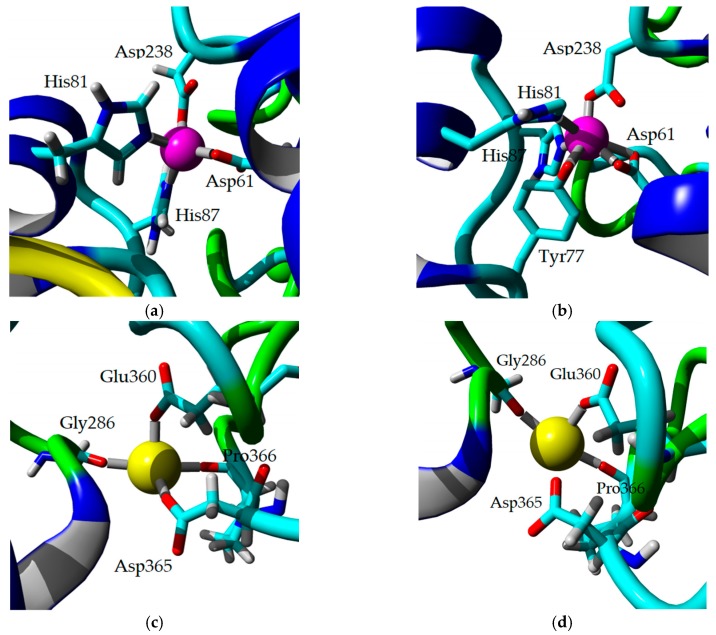
Zinc binding site and calcium-binding site in modeled structure of 5M mutant and Earth-grown T1 lipase crystal structure. (**a**) Zinc binding site in Earth-grown T1 lipase structure. The Zinc ion (Magenta sphere) are coordinated in tetrahedral form by Asp61, His81, His87, and Asp238. (**b**) Zinc binding site in 5M mutant coordinated by Asp61, Tyr77, His81, His87, and Asp238. (**c**) Calcium-binding site in Earth-grown T1 lipase structure coordinated by residues Asp365, Glu360, Gly286, and Pro366. (**d**) Calcium-binding site in 5M mutant coordinated by residues Asp365, Glu360, Gly286, and Pro366.

**Figure 10 ijms-20-02561-f010:**
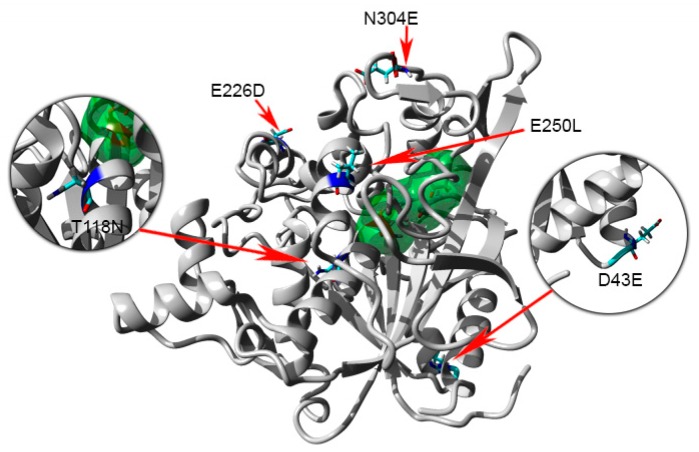
Structure of predicted structure of 5M mutant lipase with mutation location sites. The catalytic triad residues (Ser113, Asp317 and His358) are colored as green surface. Mutated residues (Glu43, Asn118, Asp226, Glu250 and Glu304) colored by element.

**Figure 11 ijms-20-02561-f011:**
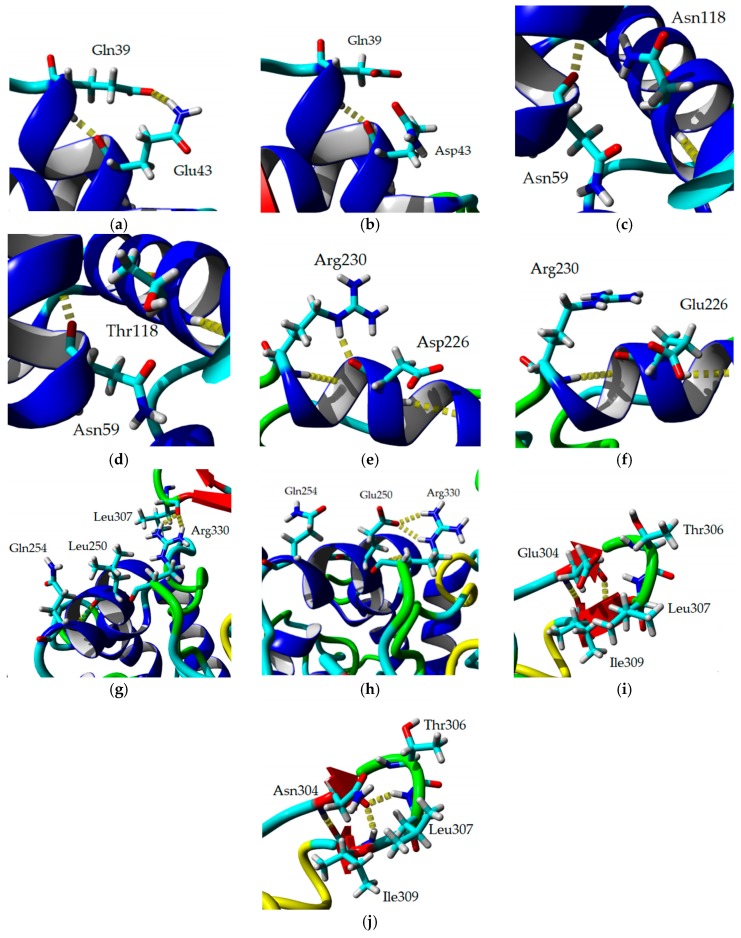
Structure model of 5M mutant and three-dimensional structure of Earth-grown T1 lipase crystal structure with hydrogen bonds and ion interaction. (**a**) Hydrogen bond between residues Gln39 and Glu43 in 5M. (**b**) Hydrogen bond between residues Gln39 and Glu43 in T1. (**c**) Hydrogen bond between residues Asn59 and Asn118 in 5M. (**d**) Hydrogen bond between residues Asn59 and Thr118 in T1. (**e**) Hydrogen bond between residues Asp226 and Arg230 in 5M. (**f**) Hydrogen bond between residues Glu226 and Arg230 in T1. (**g**) Hydrogen bond between residues Leu250, Arg330, and Gln254 in 5M. (**h**) Hydrogen bond and ion interaction between residues Glu250, Arg330, and Gln254 in T1. (**i**) Hydrogen bond between residues Glu304 and Thr306 in 5M. (**j**) Hydrogen bond between residues Glu304 and Thr306 in T1.

**Table 1 ijms-20-02561-t001:** The effect of metal ions on enzyme activity of wt-HT1 and 5M mutant lipases.

Lipase	Concentration (mM)	Na^2+^	Ca^2+^	Mg^2+^	Fe^3^	Ni^2+^	Cu^2+^	Zn^2+^
5M	1	91.5 ± 6.7	150.0 ± 2.6	59.6 ± 2.2	18.8 ± 5.9	86.4 ± 5	15.3 ± 2.3	75.0 ± 0.7
	5	76.7 ± 0.4	193.0 ± 5.0	53.7 ± 1.8	7.8 ± 2.1	78.0 ± 8.9	0.0 ± 0.0	21.8 ± 1.2
wt-HT1	1	102.6 ± 4.1	122.0 ± 1.7	81.6 ± 5.2	55.9 ± 1.3	106.0 ± 2.4	51.2 ± 4.2	78.7 ± 1.2
	5	103.3 ± 0.6	123.0 ± 4.3	101.0 ± 3.8	9.9 ± 2.9	93.0 ± 6.5	15.6 ± 4.0	25.2 ± 5.4

**Table 2 ijms-20-02561-t002:** Secondary structure distribution of wt-HT1 and 5M mutant lipases with and without calcium ion.

Lipases with and without Ca^2^⁺	α-Helix	β-Sheet	Turn	Random
wt-HT1 + Ca^2^⁺	15.30%	49.90%	7.60%	27.20%
5M + Ca^2^⁺	20.30%	46.30%	8.10%	25.30%
wt-HT1	28.70%	17.60%	22.40%	31.30%
5M	23.90%	18.30%	18.00%	39.70%

**Table 3 ijms-20-02561-t003:** Melting temperature point (T_m_) of wt-HT1 and 5M mutant lipases with and without calcium ion using Circular Dichroism.

Lipases with and without Ca^2^⁺	Melting Temperature (T_m_)
wt-HT1 + Ca^2^⁺	83.0 °C ± 0.2
5M + Ca^2^⁺	76.0 °C ± 0.7
wt-HT1	70.9 °C ± 0.1
5M	67.7 °C ± 0.8

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
