# Peer review of "Changes of Thermostability, Organic Solvent, and pH Stability in Geobacillus zalihae HT1 and Its Mutant by Calcium Ion"

_ijms, 2019, doi:10.3390/ijms20102561_

Reviewer 1 Report

The manuscript describes the effect of Ca ion in the structure of the lipase from Geobacillus zalihae.

The manuscript is mainly and formally correct but I have some doubts about the novelty of the studies and if the material is important enough for publication in this journal.

In this way, authors have to describe in intro the novelty and importance of the presented studies. In fact there are other manuscripts published by the same group with similar studies. The only difference is that  the addition of Ca is able to stabilize the enzyme structure and this was properly characterized.

The importance of this enzyme would be described in intro.

Other minor question:

The purification part, in table the purification factor is 2.5 and 3 respectively but in electrophoresis is clear that this factor have to be much higher than that attending to the figure. What is the explanation for that?. Is it a question of different dilutions? If yes, I suggest repeating it with similar concentrations in order to compare.

Author Response

The manuscript describes the effect of Ca ion in the structure of the lipase from Geobacillus zalihae.

The manuscript is mainly and formally correct but I have some doubts about the novelty of the studies and if the material is important enough for publication in this journal.

In this way, authors have to describe in intro the novelty and importance of the presented studies. In fact there are other manuscripts published by the same group with similar studies. The only difference is that  the addition of Ca is able to stabilize the enzyme structure and this was properly characterized.

Response 1:

Besides the effect of Ca in the structure of lipase, the manuscript also described about the important of organic solvent tolerant lipase which are important in industry applications. This paper is the first study to manipulate the information obtained from space crystal structure for the production of new lipase which increased in stability towards organic solvents.

Other minor question:

The purification part, in table the purification factor is 2.5 and 3 respectively but in electrophoresis is clear that this factor have to be much higher than that attending to the figure. What is the explanation for that?. Is it a question of different dilutions? If yes, I suggest repeating it with similar concentrations in order to compare.

Response 2:

As requested by Examiner, the SDS-Page has been repeat using the similar concentration of purified lipases.

Reviewer 2 Report

in the manuscript, the authors proposed a mutagenesis of a lipase enzyme in order to improve its stability. Although the overall merit in the principal idea, these mutations are not necessarily positive in the field of stability. Lipase enzymes are often used for the production of biodiesel, foods and oils. The high temperature stability is very important. I invite the authors to improve this field, as reported in your conclusion.

Author Response

in the manuscript, the authors proposed a mutagenesis of a lipase enzyme in order to improve its stability. Although the overall merit in the principal idea, these mutations are not necessarily positive in the field of stability. Lipase enzymes are often used for the production of biodiesel, foods and oils. The high temperature stability is very important. I invite the authors to improve this field, as reported in your conclusion.

Response 1:

The introduction has been updated with the information regarding the importance of organic solvent tolerant lipase in industry application. 

Reviewer 3 Report

In this study, the lipase crystal structure in space was used to design the substitution of amino acid sequence in lipase. As the result, the lipase has good thermostability, organic solvent and pH stability. The article shows that the space-grown crystal structure of lipase can effectively improve the thermal stability of lipase than the earth-grown crystal structure. I am not a space scientist and I also lack of space crystallization knowledge. Therefore, I cannot understand the relationship between space crystallization and the amino acid sequence of the lipase. The author should explain the relationship between space crystallization and the amino acid sequence in detail, otherwise please remove the mentions related to space crystallization. This study can be published with major revision and clarification of the issues. The issues are explained below:

1.      P1, l.22. How did the crystallization under space condition? Is the crystal obtained in the space station?

2.      P1, l.28. wt-HT1 is the code name. Please indicate that it is a nucleic acid, protein or microbial strain.

3.      P2, l.81. JAXA-UPM should full name.

4.      P2, l.84. How did the author know that the main differences of T1 lipase structure based on the analysis of earth and space can be summarized to its hydrogen bond and salt bridge formation? Why the space-grown lipase structure has the better activity and stability?

5.      P2, l.80-92. What is the relationship between space-grown structure, hydrogen bond, amino acid sequence and functions of calcium ion? Please explain in detail and add in the text.

6.      P3, l.98-99. The authors used the data obtained from the earlier study to design the substitution of amino acids that the earlier study should be cited. I also don`t know the simulation study of space-grown crystal structures can be used to decide the substitution of amino acids in lipase and enhanced it stability. Authors should describe in detail how to use the simulation study of space-grown crystal structures to decide the substitution of amino acids.

7.      P3, l.106. The wt-HT1 and 5M are the code name. Please indicate that are nucleic acids, enzyme or microbial strains.

8.      P3, l.109-111. The SDS page could not prove that the amino acid had been replaced in lipase wt-HT1. The authors should show evidence that the amino acid has been replaced in lipase wt-HT1. For example, amino acid or DNA sequencing. This is very needed to prove, because it can prove that authors claim the space growth crystal structure can be used to predict the amino acid substitution of lipase to increase its stability.

9.      P9,l.225. The authors used CD spectra used to confirm the structure conformation of wt-HT1 and 5M lipases (Fig. 7). The secondary structure distribution of wt-HT1 and 5M mutant lipases are shown in table 3. The CD spectra covert to secondary structure distribution does not mention and describe in section 4.9. Please add in the Materials and Methods.

10.   P11, Fig. 8. The melting point usually obtained from differential scanning calorimetry. How to get the melting point from CD spectra? Please describe in the text.

11.  P18, l.473. The activity was determined by measuring the absorbance at 715 nm. How to change the absorbance to a unit of lipase? Please provide unit conversion method.

12.  P.19, l.520-523. The author used 6 pages (p12-17) to explain and discuss the homology modelling and structural analysis of lipase. However, the authors does not mention and describe about this data how to obtained by using Yet Artificial Reality Application. The authors should add the detail about used the YASARA and how the YASARA can be obtained the structural analysis in the Materials and Methods.

Author Response

In this study, the lipase crystal structure in space was used to design the substitution of amino acid sequence in lipase. As the result, the lipase has good thermostability, organic solvent and pH stability. The article shows that the space-grown crystal structure of lipase can effectively improve the thermal stability of lipase than the earth-grown crystal structure. I am not a space scientist and I also lack of space crystallization knowledge. Therefore, I cannot understand the relationship between space crystallization and the amino acid sequence of the lipase. The author should explain the relationship between space crystallization and the amino acid sequence in detail, otherwise please remove the mentions related to space crystallization. This study can be published with major revision and clarification of the issues. The issues are explained below:

1.      P1, l.22. How did the crystallization under space condition? Is the crystal obtained in the space station? –

The crystallization of T1 lipase under space and earth conditions were done using modified capillary method. Added in the text Lane and Lane 139. For full revision, please refer to the previous report by Aris et al. (2014).

Aris, S.N.A.M., Leow, T.C., Ali, M.S.M., Mahiran, B., Salleh, A.B. and Rahman, R.N.Z.A. Crystallographic Analysis of Ground and Space Thermostable T1 Lipase Crystal Obtained via Counter Diffusion Method Approach. BioMed Research International. 2014. 1 – 8

2.      P1, l.28. wt-HT1 is the code name. Please indicate that it is a nucleic acid, protein or microbial strain.

Lane 31

3.      P2, l.81. JAXA-UPM should full name.

Lane 138

4.      P2, l.84. How did the author know that the main differences of T1 lipase structure based on the analysis of earth and space can be summarized to its hydrogen bond and salt bridge formation? Why the space-grown lipase structure has the better activity and stability?

The information regarding space structure has been updated in the text in introduction part Lane 52 – 69, Lane 145 -154

5.      P2, l.80-92. What is the relationship between space-grown structure, hydrogen bond, amino acid sequence and functions of calcium ion? Please explain in detail and add in the text.

Has been updated in the introduction as suggested by reviewer. Lane 52 – 69, Lane 145 -154

6.      P3, l.98-99. The authors used the data obtained from the earlier study to design the substitution of amino acids that the earlier study should be cited. I also don`t know the simulation study of space-grown crystal structures can be used to decide the substitution of amino acids in lipase and enhanced it stability. Authors should describe in detail how to use the simulation study of space-grown crystal structures to decide the substitution of amino acids.

The previous study has been cited in the text as suggested.

The application of molecular dynamics simulation for amino acid substitution has been describe in introduction as suggested by reviewer. Lane 70 – 88, Lane 145 -154

7.      P3, l.106. The wt-HT1 and 5M are the code name. Please indicate that are nucleic acids, enzyme or microbial strains.

Already describe in methods Lane 505. But as suggested by reviewer I mention again in Lane 164.

8.      P3, l.109-111. The SDS page could not prove that the amino acid had been replaced in lipase wt-HT1. The authors should show evidence that the amino acid has been replaced in lipase wt-HT1. For example, amino acid or DNA sequencing. This is very needed to prove, because it can prove that authors claim the space growth crystal structure can be used to predict the amino acid substitution of lipase to increase its stability.

The amino acid alignment already provided in Figure.  But as suggested by reviewer I added the result of DNA sequence alignment in supplementing result in Figure S1.

9.      P9,l.225. The authors used CD spectra used to confirm the structure conformation of wt-HT1 and 5M lipases (Fig. 7). The secondary structure distribution of wt-HT1 and 5M mutant lipases are shown in table 3. The CD spectra covert to secondary structure distribution does not mention and describe in section 4.9. Please add in the Materials and Methods.

As suggested, the sentence was added in the text. The analysis of structural element was determined using Spectra ManagerTM Suite Software (JASCO, Japan). Page 20, Lane 600.

10.    P11, Fig. 8. The melting point usually obtained from differential scanning calorimetry. How to get the melting point from CD spectra? Please describe in the text.

As suggested by reviewer, the method has been updated. Lane 608.

11.    P18, l.473. The activity was determined by measuring the absorbance at 715 nm. How to change the absorbance to a unit of lipase? Please provide unit conversion method.

As suggested by reviewer, the equation has been added in the text. Lane 553.

12.    P.19, l.520-523. The author used 6 pages (p12-17) to explain and discuss the homology modelling and structural analysis of lipase. However, the authors does not mention and describe about this data how to obtained by using Yet Artificial Reality Application. The authors should add the detail about used the YASARA and how the YASARA can be obtained the structural analysis in the Materials and Methods.

The section 5.0 in Materials and methods has been updated as suggested.

Round  2

Reviewer 1 Report

I still not convinced about figure 2 compared with table 1. In this way, viewing the figure c??

this is a bizarre result. Using the program just quantify, lane analysis, the density of the color is 1144.32 for M, 8037 for C and 632.66 for 1. This is in acordance with previous commentary and means around 12 fold purification factor. This SDS-PAGE can not be compared and i consider this as an error. If the authors asume 3 the purification factor, this is a wrong figure.

Reviewer 3 Report

Accepted

Author Response

Dear Reviewer,

Thank you for your valuable comment and suggestion. 

Round  3

Reviewer 1 Report

In spite of i am not specially agree with the purification data (factor of 2 is difficult to asume considering the SDS-PAGE) this is not the most relevant question of the manuscript thus, i recommend its publication